# Associations of cardiovascular health and social determinants of health with the risks of all-cause and cause-specific mortality

**Boxuan Pu** [1]*, **Zun Wang**[2]

**1** National Clinical Research Center for Cardiovascular Diseases, Fuwai Hospital, National Center for Cardiovascular Diseases, Chinese Academy of Medical Sciences and Peking Union Medical College, Beijing, China, **2** Youanmen Community Healthcare Center of Fengtai District, Beijing, China

* puboxuan@fuwai.com

## Abstract

### Background

While cardiovascular health (CVH) and social determinants of health (SDoH) are independently associated with mortality, their combined effects on mortality remain unclear. The study aimed to examine the mediating, interacting, and combined effects of CVH and SDoH on mortality risks.

### Methods

We analyzed data from 20,096 adults in the National Health and Nutrition Examination Survey 2007–2018. CVH was assessed using the Life's Essential 8 (LE8) score and categorized into low, moderate, or high. Cumulative unfavorable SDoH burden was assessed and divided into low and high burdens of unfavorable SDoH. Multivariate Cox regression models were used to analyze the combined associations of SDoH and CVH with mortality. Mediation analysis was used to analyze the mediating role of CVH between SDoH and mortality. Interaction effects were tested by adding interaction terms between CVH and SDoH in the Cox models.

### Results

A high burden of unfavorable SDoH was associated with higher risks of all-cause, cardiovascular disease (CVD), and cancer mortality. CVH significantly mediated these associations, accounting for 16.70% for all-cause mortality, 22.22% for CVD mortality, and 17.70% for cancer mortality. A significant additive interaction between SDoH and CVH was observed for all-cause mortality. Compared with individuals exhibiting neither risk factor, those with both low CVH and a high burden of unfavorable SDoH had significantly elevated risks of all-cause (hazard ratio [HR]: 4.60; 95% confidence interval [95% CI]: 2.87–7.37), CVD (2.80; 1.10–7.10), and cancer mortality (6.10; 2.13–17.50).

**Data availability statement:** Data availability statement The datasets generated and analyzed in this study are available in the NHANES website, https://wwwn.cdc.gov/nchs/nhanes/.

**Funding:** The author(s) received no specific funding for this work.

**Competing interests:** The authors have declared that no competing interests exist.

## Conclusions

The coexistence of low CVH and a high burden of unfavorable SDoH was associated with increased risks of all-cause, CVD, and cancer mortality. These findings have implications for integrating SDoH and CVH in clinical practice and public health strategies to improve survival.

## Introduction

Growing evidence has highlighted the crucial role of social determinants of health (SDoH) in causing disparities in clinical outcomes and health equity [1]. The World Health Organization defines SDoH as "the conditions in which people are born, grow, live, work, and age, and the wider set of forces and systems shaping the condition of daily life" [2]. The U.S. Healthy People 2030 initiative identifies a thorough SDoH framework covering five key domains: economic stability, education access and quality, health care access and quality, neighborhood and built environment, and social community context [3,4]. Adverse SDoH have been associated with high burdens of cardiovascular risk factors and adverse outcomes, such as cardiovascular disease (CVD), cancer, and mortality [1,5–7]. Approximately 30–55% of health outcomes are attributable to SDoH, suggesting their role as primary drivers of preventable health disparities [8]. Thus, public health interventions tackling adverse SDoH are critical for reducing health disparities.

The cumulative burden of adverse SDoH is associated with a lower likelihood of achieving optimal cardiovascular health (CVH) [9]. thereby increasing risks of mortality and CVD. SDoH provide the daily context for CVH and influence the potential to optimize CVH, as well as the effectiveness of interventions aiming at improving it [10]. Therefore, it is plausible that the association between SDoH and mortality could be mediated by CVH. However, this relationship remains unknown. In 2010, the American Heart Association (AHA) first proposed Life's Simple 7 to measure CVH using 7 key metrics: smoking, diet, physical activity, body mass index (BMI), blood pressure (BP), cholesterol, and fasting glucose [11]. The metric was updated in 2022 as Life's Essential 8 (LE8), adding sleep and refining other domains [12]. High CVH has been demonstrated to be related to longer life expectancy and reduced premature mortality [13,14]. Several studies explored the combined effects of specific CVH components and specific SDoH domains on health outcomes [15–17]. However, these studies mainly examined one or two specific domains of SDoH or lifestyle components of CVH, and did not fully examine the combined effects of overall SDoH and CVH. Despite existing knowledge on the independent relationship of CVH and SDoH with health outcomes and their interlinkages, it remains unclear whether there is a combined effect of CVH and SDoH on mortality risks and to what extent CVH mediates the SDoH-mortality association.

To address these knowledge gaps, we analyzed data from the 2007–2018 National Health and Nutrition Examination Survey (NHANES) to investigate whether CVH modifies the association between SDoH and mortality. Further, this study also aimed to evaluate the interaction or combined associations of SDoH and CVH with mortality risks in a large representative sample of U.S. adults.

## Methods

### Study participants

In a two-year cycle starting at 1999–2000, NHANES used a stratified, multistage probability design to collect cross-sectional health and nutrition data from a nationally representative sample of the noninstitutionalized U.S. civilian population. Participants underwent an in-home interview followed by a visit to a mobile screening facility for anthropometric measurements, physiological examinations, and laboratory testing. The National Center for Health Statistics (NCHS) Ethics Review Board approved the NHANES protocols (No.: #2005–06, #2011–17, #2018-01), and all participants provided informed consent. This study used data from 6 consecutive NHANES cycles spanning 2007–2018. We excluded participants aged <20 years (n = 25,072), pregnant individuals (n = 372), those without mortality data (n = 115), and individuals with missing data on any individual component for SDoH (n = 1,809) or CVH (n = 12,378). Finally, the current study included 20,096 participants (S1 Fig).

### CVH assessment

CVH was assessed using the LE8 score, which includes 4 health behaviors (diet, physical activity, tobacco/nicotine exposure, and sleep) and 4 health factors (BMI, non-high-density lipoprotein [non-HDL] cholesterol, blood glucose, and BP) (S1 Table). Diet quality was calculated using the Healthy Eating Index-2015 based on 24-hour dietary recalls [18]. Self-reported questionnaires provided data on sleep duration, smoking status, and physical activity levels. Trained investigators measured BP, blood glucose, height, and weight. BMI was calculated as weight (kg)/ height (m$^2$). Glycated hemoglobin was measured using high-performance liquid chromatography methods. Non-HDL cholesterol was calculated as total cholesterol minus HDL cholesterol. Each LE8 component is scored 0–100, and the total CVH score represents the unweighted average of all 8 metrics, ranging from 0–100, with higher scores indicating better CVH. The CVH was categorized into low (LE8 < 50), moderate (50 ≤ LE8 < 80), or high (LE8 ≥ 80) [12,13].

### SDoH assessment

By the U.S. Healthy People 2030, this study identified 8 sub-items of SDoH across 5 domains: social and community context (marital status), neighborhood and built environment (home ownership), health care access and quality (health insurance coverage and type), education access and quality (education level), and economic stability (employment status, family poverty-income ratio, and food security) (S2 Table) [3]. Each item was dichotomized using conventional cutoff points as favorable (level 0) or unfavorable (level 1). The cumulative burden of unfavorable SDoH was calculated by summing unfavorable items. Participants were categorized into high burden (>2 unfavorable SDoH) or low burden (≤2 unfavorable SDoH) based on previous studies [19,20].

### Covariates

Covariates were obtained via standardized questionnaires, including self-reported age, sex, race/ethnicity, cancer history, and CVD history. The categories of race/ethnicity included Mexican American, non-Hispanic Black, non-Hispanic White, and other races. CVD history was defined as a self-reported physician diagnosis of any of the following: congestive heart failure, coronary heart disease, angina, myocardial infarction, or stroke. Likewise, cancer history was defined as a self-reported physician diagnosis of cancer or malignancy.

### Outcomes

Deaths were ascertained through linkage with the National Death Index records, with follow-up through December 31, 2019. Underlying causes of death were determined using codes from the International Classification of Diseases, 10th

Revision. CVD mortality was defined as death due to heart diseases (I00 to I09, I11, I13, I20 to I51, I60 to 169), and cancer mortality was defined as death due to cancer (C00 toC97).

## Statistical analyses

We used weighted mean (standard error [SE]) to present continuous variables, and counts (survey-weighted percentages) for categorical variables. To compare characteristics across different levels of SDoH or CVH, we used analysis of variance for continuous variables and the chi-square test for categorical variables. The relationship between SDoH and CVH was evaluated using linear regression analysis.

Multivariate Cox proportional hazards models were used to estimate risks of all-cause and cause-specific mortality in relation to SDoH or CVH. The models were adjusted for age, sex, race/ethnicity, CVD, and cancer history. Mediation analyses were conducted using the R package "mediation" to quantify the extent to which CVH mediates the association between SDoH and mortality, with the proportion of mediation quantified by comparing models with and without adjustment for CVH. Bootstrap resampling (1000 simulations) was used to estimate of the mediation effect and its 95% confidence interval (95% CIs). Additionally, model predictive performance was evaluated by calculating the concordance index (C-index), integrated discrimination improvement (IDI), and net reclassification index (NRI) for models with and without SDoH.

A stratified analysis was performed based on SDoH levels to examine the association between CVH and mortality within different SDoH subgroups. To assess interactions between SDoH and CVH, an interaction term was incorporated into the multivariate models. Additive interactions were assessed via relative excess risk (RERI), attributable proportion (AP), and synergy index (SI). Multiplicative interactions were tested using interaction terms in Cox models. For the combined analyses, participants were classified into six groups by combined CVH and SDoH levels, and mortality risks were compared across these categories, with high CVH and a low burden of unfavorable SDoH as reference.

Subgroup analyses were conducted to assess potential effect modifications by age (<65 vs. ≥ 65 years), sex (men vs. women), and race/ethnicity (White vs. non-White). To evaluate the robustness of the primary findings, three sensitivity analyses were performed: first, excluding participants who died within the first two years of follow-up to minimize reverse causality; second, excluding those with CVD or cancer at baseline; and third, further adjusting for additional covariates including drinking status, chronic kidney disease, and chronic obstructive pulmonary disease.

No missing data were observed for covariates. All analyses were conducted using R 4.2.2 while accounting for the complex survey design of the NHANES database, including sampling weights, clustering, and stratification, ensuring nationally representative estimates. Two-sided $P$ values <0.05 considered statistically significant.

## Results

### Participant characteristics

Among 20,096 participants (weighted population: 146,521,183), the mean age (SE) was 47.83 (0.28) years, and 10,254 (51.38%) were women. Of these participants, 11,262 (57.26%) had a high burden of unfavorable SDoH. Compared to the low burden group, those with a high burden of unfavorable SDoH were younger, more likely to be women and non-White, and had higher prevalence of CVD and poor CVH, and have poor CVH. Additionally, 5,275 (22.51%) individuals had low CVH, and 1,623 (10.76%) had high CVH. Individuals with high CVH tended to be younger, more likely to be women and White, and had lower prevalences of both CVD and cancer than those with low CVH (Table 1). Participants excluded from the analysis were younger, more likely to be non-White, and had a lower proportion of participants with CVD or cancer (S3 Table).

### Mediation of CVH on SDoH-mortality association

During a median follow-up of 6.8 years (interquartile range: 4.0–9.9), 1,774 participants (6.29%) died from all causes, including 518 (1.75%) from CVD and 451 (1.65%) from cancer. After multivariable adjustment, individuals with a high

**Table 1. Baseline characteristics of participants.**

| Characteristics | Total | Unfavorable SDoH | | | CVH | | | |
|---|---|---|---|---|---|---|---|---|
| | | Low burden | High burden | *P* value | Low | Moderate | High | *P* value |
| Weighted N (weighted %) | 146,521,183 (100.0) | 62,628,270 (42.74) | 83,892,914 (57.26) | | 32,979,500 (22.51) | 97,782,843 (66.74) | 15,758,841 (10.76) | |
| No. of participants in sample | 20096 | 8834 | 11262 | | 5275 | 13189 | 1632 | |
| Age, years (SE) | 47.83 (0.28) | 49.51 (0.28) | 45.59 (0.41) | <0.001 | 52.89 (0.34) | 47.50 (0.31) | 39.32 (0.55) | <0.001 |
| Women, n (weighted %) | 10254 (51.38) | 4302 (49.94) | 5952 (53.51) | <0.001 | 2526 (47.90) | 6718 (50.57) | 1010 (63.71) | <0.001 |
| Race/ethnicity, n (weighted %) | | | | <0.001 | | | | <0.001 |
| Mexican | 2863 (7.78) | 760 (3.89) | 2103 (3.74) | | 699 (7.79) | 1989 (8.18) | 175 (5.27) | |
| White | 9098 (70.18) | 4858 (80.34) | 4240 (56.58) | | 2381 (68.58) | 5870 (69.79) | 847 (76.02) | |
| Black | 4000 (9.84) | 1392 (5.96) | 2608 (15.02) | | 1364 (13.30) | 2476 (9.52) | 160 (4.53) | |
| Other | 4135 (12.20) | 1824 (9.81) | 2311 (15.41) | | 831 (10.33) | 2854 (12.52) | 450 (14.18) | |
| Medical history, n (weighted %) | | | | | | | | |
| CVD history | 2144 (8.21) | 679 (6.30) | 1465 (10.78) | <0.001 | 969 (15.49) | 1131 (6.69) | 44 (2.45) | <0.001 |
| Cancer history | 1972 (10.40) | 983 (11.57) | 989 (8.84) | <0.001 | 581 (11.71) | 1300 (10.52) | 91 (6.92) | <0.001 |
| SDoH, n (weighted %) | | | | | | | | |
| Unemployed | 7714 (31.26) | 2177 (22.16) | 5537 (43.43) | <0.001 | 2727 (43.95) | 4654 (29.03) | 333 (18.48) | <0.001 |
| Family income-to-poverty ratio<300% | 12535 (48.78) | 2232 (19.43) | 10293 (88.09) | <0.001 | 3934 (63.25) | 7913 (46.48) | 688 (32.72) | <0.001 |
| Marginal or lower food security | 6232 (23.13) | 513 (4.67) | 5719 (47.86) | <0.001 | 2181 (33.92) | 3778 (21.29) | 273 (11.97) | <0.001 |
| Not owning a home | 7435 (30.70) | 1102 (11.64) | 6333 (56.23) | <0.001 | 2165 (34.39) | 4700 (29.71) | 570 (29.14) | 0.003 |
| Less than high school | 4363 (13.85) | 401 (3.41) | 3962 (27.83) | <0.001 | 1652 (23.07) | 2607 (12.40) | 104 (3.53) | <0.001 |
| No regular health care access | 3608 (16.67) | 513 (6.45) | 3095 (30.37) | <0.001 | 867 (16.68) | 2422 (16.66) | 319 (16.75) | 0.996 |
| No private health insurance | 9283 (35.71) | 1057 (10.53) | 8226 (69.44) | <0.001 | 3020 (48.01) | 5837 (34.14) | 426 (19.71) | <0.001 |
| Not married or living with a partner | 7938 (35.29) | 1689 (18.49) | 6249 (57.79) | <0.001 | 2268 (38.38) | 5042 (34.46) | 628 (33.95) | 0.006 |
| AHA LE8 score, mean (SE) | | | | | | | | |
| Total CVH score | 61.03 (0.32) | 63.61 (0.35) | 57.58 (0.37) | <0.001 | 41.24 (0.17) | 63.82 (0.15) | 85.15 (0.12) | <0.001 |
| HEI-2015 diet score | 39.01 (0.24) | 40.77 (0.27) | 36.65 (0.27) | <0.001 | 33.57 (0.32) | 39.32 (0.24) | 48.42 (0.59) | <0.001 |
| Physical activity score | 46.36 (0.83) | 53.10 (1.00) | 37.34 (0.90) | <0.001 | 11.59 (0.58) | 50.28 (0.78) | 94.83 (0.45) | <0.001 |
| Tobacco/nicotine exposure score | 72.13 (0.55) | 79.12 (0.53) | 62.77 (0.80) | <0.001 | 47.11 (0.84) | 76.77 (0.53) | 95.74 (0.38) | <0.001 |
| Sleep health score | 70.04 (1.02) | 73.32 (1.47) | 65.65 (0.91) | <0.001 | 47.44 (1.61) | 73.68 (0.98) | 94.77 (0.39) | <0.001 |
| Body mass index score | 60.32 (0.42) | 61.17 (0.53) | 59.19 (0.53) | 0.007 | 38.64 (0.62) | 62.76 (0.43) | 90.59 (0.55) | <0.001 |
| Blood lipid score | 63.43 (0.38) | 62.73 (0.45) | 64.37 (0.44) | 0.010 | 44.51 (0.61) | 65.81 (0.41) | 88.28 (0.67) | <0.001 |
| Blood glucose score | 79.61 (0.31) | 80.68 (0.41) | 78.19 (0.37) | <0.001 | 63.29 (0.47) | 82.41 (0.30) | 96.46 (0.36) | <0.001 |
| Blood pressure score | 57.35 (0.27) | 57.99 (0.33) | 56.50 (0.37) | 0.002 | 43.80 (0.51) | 59.54 (0.27) | 72.15 (0.26) | <0.001 |

Data are survey-weighted mean (SE) or N (weight percentage %).

Abbreviations: SE: standard error; CVD: cardiovascular diseases; SDoH: social determinants of health; CVH: cardiovascular health; AHA: American Heart Association; LE8: Life's Essential 8; HEI-2015: healthy eating index-2015.

burden of unfavorable SDoH had higher risks of all-cause (hazard ratio [HR], 2.32; 95% CI: 2.04–2.63), CVD (HR, 2.21; 95% CI: 1.75–2.80), and cancer mortality (HR, 1.72; 95% CI: 1.34–2.22) compared to the low-burden group. Similarly, individuals in the low CVH group had higher risks of all-cause (HR, 2.72; 95% CI: 1.85–3.99), CVD (HR, 2.84; 95% CI: 1.15–6.97), and cancer mortality (HR, 2.56; 95% CI: 1.21–5.43) compared to those in the high CVH group (Table 2). Furthermore, each unit increase in unfavorable SDoH correlated with significantly lower CVH scores (β=−1.91; 95% CI: −2.00- −1.81, *P*<0.001) (S4 Table).

**Table 2. Independent association of social determinants of health and cardiovascular health with all-cause and cause-specific mortality.**

| Outcomes | Death/No. | Weighted death (%) | Adjusted HR (95% CI) | Adjusted *P* value |
|---|---|---|---|---|
| **All-cause mortality** | | | | |
| Unfavorable SDoH | | | | |
| Low burden | 554/8834 | 3,654,817 (4.36) | 1 (Reference) | |
| High burden | 1220/11262 | 5,561,140 (8.88) | 2.32 (2.04-2.63) | <0.001 |
| Per 1 unfavorable SDoH increase | | | 1.32 (1.28-1.37) | <0.001 |
| CVH | | | | |
| High | 40/1632 | 256,500 (1.63) | 1 (Reference) | |
| Moderate | 971/13189 | 5,033,088 (5.15) | 1.53 (1.06-2.20) | 0.022 |
| Low | 763/5275 | 3,926,370 (11.91) | 2.72 (1.85-3.99) | <0.001 |
| Per 10 CVH scores decrease | | | 1.32 (1.23-1.39) | <0.001 |
| **CVD mortality** | | | | |
| Unfavorable SDoH | | | | |
| Low burden | 158/8834 | 983,648 (1.17) | 1 (Reference) | |
| High burden | 360/11262 | 1,584,263 (2.53) | 2.21 (1.75-2.80) | <0.001 |
| Per 1 unfavorable SDoH increase | | | 1.30 (1.22-1.39) | <0.001 |
| CVH | | | | |
| High | 8/1632 | 57,143 (0.36) | 1 (Reference) | |
| Moderate | 287/13189 | 1,400,529 (1.43) | 1.65 (0.67-4.04) | 0.300 |
| Low | 223/5275 | 1,110,239 (3.37) | 2.84 (1.15-6.97) | 0.023 |
| Per 10 CVH scores decrease | | | 1.37 (1.23-1.51) | <0.001 |
| **Cancer mortality** | | | | |
| Unfavorable SDoH | | | | |
| Low burden | 170/8834 | 1,180,251 (1.41) | 1 (Reference) | |
| High burden | 281/11262 | 1,248,140 (1.99) | 1.72 (1.34-2.22) | <0.001 |
| Per 1 unfavorable SDoH increase | | | 1.18 (1.09-1.28) | <0.001 |
| CVH | | | | |
| High | 11/1632 | 69,072 (0.44) | 1 (Reference) | |
| Moderate | 259/13189 | 1,447,269 (1.48) | 1.66 (0.79-3.52) | 0.200 |
| Low | 181/5275 | 912,050 (2.77) | 2.56 (1.21-5.43) | 0.014 |
| Per 10 CVH scores decrease | | | 1.20 (1.09-1.33) | <0.001 |

Multivariate models were adjusted for age, sex, race/ethnicity, cardiovascular disease history, and cancer history.

Abbreviations: SDoH: social determinants of health; CVH: cardiovascular health; HR: hazard ratio; CI: confidence interval; CVD: cardiovascular diseases.

When additionally adjusting for CVH, the HRs for the association between SDoH and all-cause, CVD, and cancer mortality were attenuated but still significant compared with models without CVH adjustment. Mediation analyses further showed that CVH significantly mediated the association between SDoH and all-cause mortality (mediation proportion: 16.70%; 95% CI: 10.90–18.96), CVD mortality (22.22%; 95% CI: 8.63–27.98), and cancer mortality (17.70%; 95% CI: 9.48–42.79) (Table 3, S2 Fig). Furthermore, incorporating SDoH significantly improved mortality risk prediction across all outcomes, with increased C-index, NRI, and IDI improvements (S5 Table).

**Interaction test and combined associations of SDoH and CVH with mortality**

A significant additive interaction between SDoH and CVH was observed for all-cause mortality (RERI, 0.42; 95% CI: 0.06–0.78; AP, 0.15; 95% CI: 0.03–0.27; SI, 1.30; 95% CI: 1.02–1.67). Approximately 15% of all-cause mortality risk was attributable to the additive interaction, suggesting the combined effects exceeded the sum of their independent effects.

**Table 3. Mediation of cardiovascular health on the associations between social determinants of health and all-cause and cause-specific mortality.**

| Outcomes | HR (95%CI) | | ACME (95%CI) | ADE (95%CI) | Total mediation effect (95%CI) | Mediation proportion (95%CI) |
|---|---|---|---|---|---|---|
| | Unadjusted for CVH | Adjusted for CVH | | | | |
| **All-cause mortality** | | | | | | |
| Low burden of unfavorable SDoH | 1 (Reference) | 1 (Reference) | 0.004 (0.003-0.005) | 0.018 (0.016-0.022) | 0.021 (0.020-0.026) | 16.70 (10.90-18.96) |
| High burden of unfavorable SDoH | 2.32 (2.04-2.63) | 2.14 (1.88-2.44) | | | | |
| **CVD mortality** | | | | | | |
| Low burden of unfavorable SDoH | 1 (Reference) | 1 (Reference) | 0.001 (0-0.002) | 0.005 (0.004-0.006) | 0.006 (0.005-0.007) | 22.22 (8.63-27.98) |
| High burden of unfavorable SDoH | 2.21 (1.75-2.80) | 2.05 (1.60-2.63) | | | | |
| **Cancer mortality** | | | | | | |
| Low burden of unfavorable SDoH | 1 (Reference) | 1 (Reference) | 0.001 (0-0.002) | 0.004 (0.002-0.005) | 0.005 (0.003-0.006) | 17.70 (9.48-42.79) |
| High burden of unfavorable SDoH | 1.72 (1.34-2.22) | 1.62 (1.26-2.09) | | | | |

Multivariate models were adjusted for age, sex, race/ethnicity, cardiovascular disease history, and cancer history.

Abbreviations: SDoH: social determinants of health; CVH: cardiovascular health; HR: hazard ratio; CI: confidence interval; CVD: cardiovascular diseases; ACME: average causal mediation effect; ADE: average direct effect.

No significant multiplicative interaction was observed (*P* for interaction>0.05) (Table 4). Stratified by SDoH burden, high CVH attenuated the increased risks of all-cause (HR, 0.43; 95% CI: 0.24–0.75) and CVD (HR, 0.08; 95% CI: 0.01–0.61) mortality associated with a high burden of unfavorable SDoH. Whether in individuals with a high or low burden of unfavorable SDoH, high CVH was both associated with lower risks of all-cause and CVD mortality, whereas the associations were stronger in the high burden group (S3 Fig).

Approximately 12.95% of participants exhibited both low CVH and a high burden of unfavorable SDoH, while only 7.81% maintained high CVH with a low burden of unfavorable SDoH (S6 Table). Individuals with low CVH and a high burden of unfavorable SDoH had the highest risks of all-cause, CVD, and cancer mortality. Specifically, compared to individuals with high CVH and a low burden of unfavorable SDoH, the HRs for all-cause, CVD, and cancer mortality in those with low CVH and a high burden of unfavorable SDoH were 4.60 (95% CI: 2.87–7.37), 2.80 (95% CI: 1.10–7.10), and 6.10 (95% CI: 2.13–17.50), respectively (Fig 1).

## Subgroup and sensitivity analyses

Subgroup analyses showed that the combined associations of SDoH and CVH with mortality risks were consistent across age, sex, and racial subgroups (all *P* for interaction>0.05, S7 Table). Sensitivity analyses yielded consistent results with

**Table 4. Additive and multiplicative interactions of social determinants of health and cardiovascular health for mortality.**

| Outcomes | Additive interaction | | | Multiplicative interaction | |
|---|---|---|---|---|---|
| | RERI (95%CI) | AP (95%CI) | SI (95%CI) | HR (95%CI) | *P* for interaction |
| All-cause mortality | 0.42 (0.06-0.78) | 0.15 (0.03-0.27) | 1.30 (1.02-1.67) | 1.18 (0.91-1.53) | 0.223 |
| CVD mortality | 0.09 (−0.58-0.76) | 0.03 (−0.22-0.29) | 1.06 (0.68-1.65) | 1.30 (0.85-2.03) | 0.091 |
| Cancer mortality | 0.60 (0.02-1.19) | 0.27 (0.03-0.51) | 1.93 (0.84-4.42) | 1.20 (0.67-2.14) | 0.546 |

All models were adjusted for age, sex, race/ethnicity, cardiovascular disease history, and cancer history. Additive interaction was evaluated using the RERI, AP, and SI between SDoH and CVH. The RERI and AP were statistically significant when their 95% CIs did not include 0, and the SI was significant when its 95% CI did not include 1. Multiplicative interaction was evaluated using HR for the interaction term between SDoH and CVH, and the multiplicative interaction was statistically significant when its 95% CI did not include 1.

Abbreviations: SDoH: social determinants of health; CVH: cardiovascular health; CI: confidence interval; CVD: cardiovascular diseases; RERI: relative excess risk due to interaction; AP: attributable proportion; SI: synergy index.

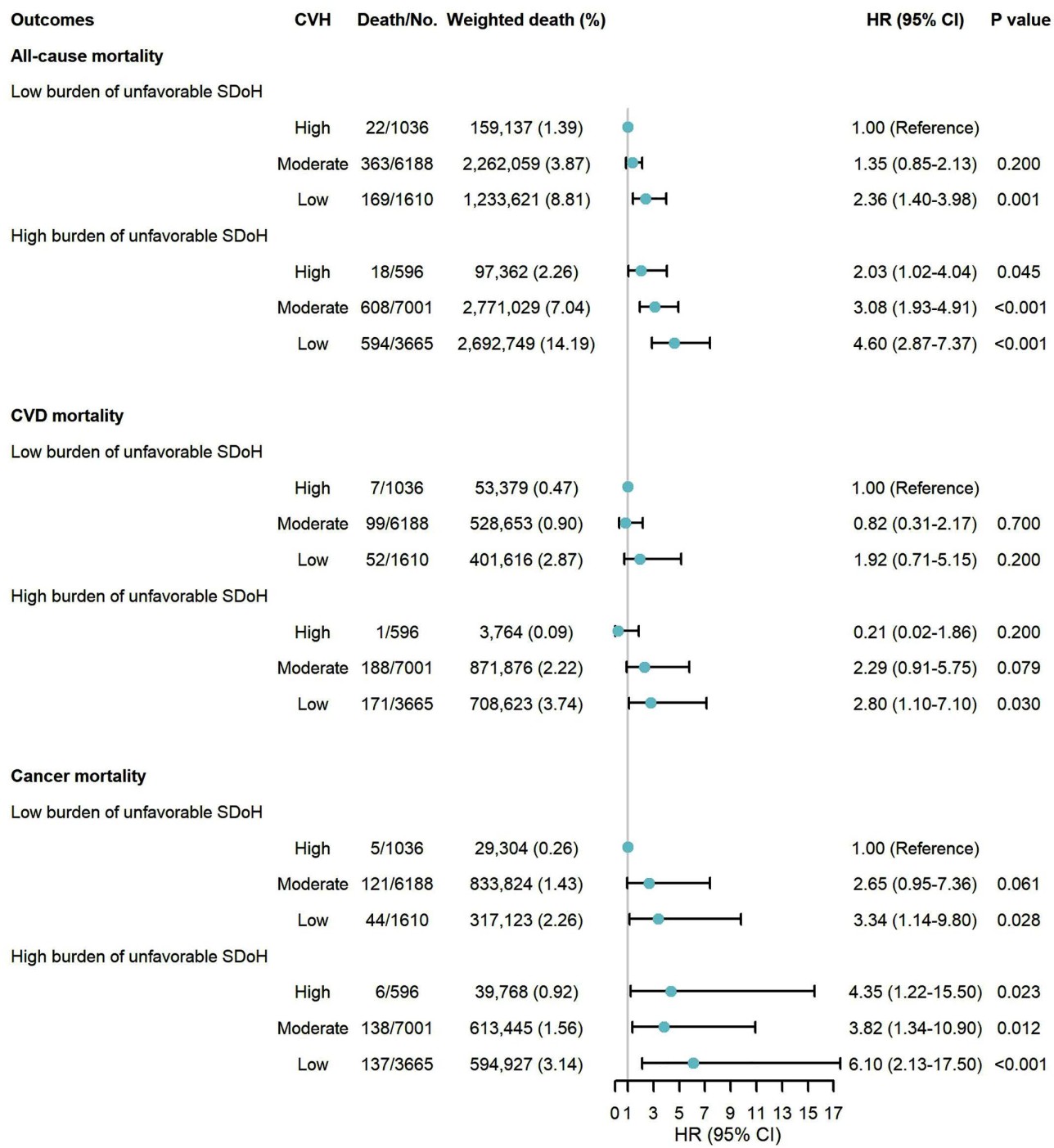

**Fig 1. Combined associations of social determinants of health and cardiovascular health with all-cause and cause-specific mortality.** Participants were classified into six groups according to the combined categories of unfavorable SDoH (low and high burden) and CVH (low, moderate, and high). Multivariate models were adjusted for age, sex, race/ethnicity, cardiovascular disease history, and cancer history. Abbreviations: SDoH: social determinants of health; CVH: cardiovascular health; HR: hazard ratio; CI: confidence interval; CVD: cardiovascular diseases.

the primary findings after excluding individuals dead within the first two years of follow-up (S8 Table), those with CVD or cancer history (S9 Table), or further adjusted for CKD, COPD, and drinking status (S10 Table).

## Discussion

This study is the first to examine the mediating, interacting, and combined effects of SDoH and CVH on all-cause and cause-specific mortality in a nationally representative sample of U.S. adults. Our findings showed that a high burden of unfavorable SDoH was associated with increased risks of all-cause, CVD, and cancer mortality. Furthermore, CVH mediated 3.92% of the association between SDoH and all-cause mortality. We observed a significant additive interaction between SDoH and CVH for all-cause mortality. Higher CVH levels attenuated the increased all-cause and CVD mortality risks associated with high unfavorable SDoH burden. In combined analyses, individuals with both a high burden of unfavorable SDoH and low CVH exhibited the highest risks of all-cause, CVD, and cancer mortality.

Our findings corroborate established evidence linking adverse SDoH to suboptimal CVH [9]. Socioeconomic advantages, including higher income, educational attainment, occupational status, and social status, are associated with high CVH [9,21,22]. SDoH plays a fundamental role in optimizing CVH, as the various domains of upstream SDoH interact to exert cumulative downstream effects on CVH through complex ways [1,7]. Our results align with the recognized association between adverse SDoH and excess mortality risk [5], with growing evidence indicating that SDoH serve as upstream drivers of the observed mortality disparities [23]. The definitions of SDoH and CVH used in our study are consistent with previous NHANES research [5,13,24], enhancing comparability and reliability of our findings. Although specific definitions of SDoH and CVH may vary across studies due to data availability, consistent evidence indicates that worse SDoH are related to worse CVH, and both are significant mortality risk factors [25,26]. While the complex relationship between SDoH and CVH on mortality remains unclear. Following adjustment for CVH, we observed an attenuation in the association between SDoH and mortality, suggesting a potential mediating role of CVH. Formal mediation analysis confirmed that CVH serves as a statistically significant but modest mediator in the pathways linking SDoH to both all-cause and cause-specific mortality. Nevertheless, the limited magnitude of mediation indicates that substantial reductions of adverse SDoH could not be achieved through promoting CVH alone, and other measures to tackle SDoH are still needed.

Moreover, we observed a significant additive interaction between SDoH and CVH on all-cause mortality. Approximately one-eighth of the excess mortality risk observed in individuals with both low CVH and high SDoH burden was attributable to synergistic effects. This suggests public health interventions simultaneously addressing both risk factors might yield greater mortality risk reduction than targeting either alone. Physiologically, adverse SDoH (e.g., poverty and limited health-care access) may induce chronic physiological stress, such as dysregulated HPA axis and increased inflammation [7], which increases the adverse effects of low CVH. Resource constraints inherent to unfavorable SDoH concurrently limit the ability to promote CVH behaviors, such as accessing healthy food, safe environments for physical activity, and preventive care [10,23], thus creating a vicious cycle that accelerates health decline. However, the result of our study differs from a Chinese cohort study reporting no interaction between CVH and SDoH [27]. This discrepancy may result from differences in SDoH and CVH definitions, as they used only 5 SDoH sub-items and did not use LE8 to assess CVH. Furthermore, it is noteworthy that high CVH provided greater protection against all-cause and CVD mortality in individuals facing SDoH disadvantages, suggesting the probability of optimizing CVH to mitigate mortality disparities exacerbated by adverse SDoH.

The current study provides novel findings regarding the combined impacts of SDoH and CVH on mortality risks in U.S. adults. Our findings align with the Chinese cohort study in community-dwelling adults without CVD, demonstrating that a high burden of unfavorable SDoH and poor CVH had a combined effect on major adverse cardiovascular events and all-cause mortality [27]. Nationwide cohort studies in the U.K. and the U.S. also reported elevated mortality and CVD risks among individuals with low socioeconomic status (SES) and few healthy lifestyles [15]. While previous research examined combined effects of one or two domains of SDoH and lifestyle components of CVH on clinical outcomes

[28,29], our assessment across all SDoH domains and AHA's standardized LE8 provides a comprehensive evaluation of the combined effects of those two risk factors. This approach enables identification of high-risk subgroups, particularly those experiencing dual burden of adverse SDoH and suboptimal CVH, who face elevated mortality risks. Despite existing evidence of SDoH's prognostic significance, most mortality prediction frameworks often overlook the contribution of SDoH in risk assessment [30,31]. Our predictive modeling showed that incorporating SDoH and CVH metrics improved risk stratification and prediction accuracy beyond models without SDoH, probably enabling early identification of vulnerable populations and optimized preventive interventions. Nevertheless, large-scale prospective cohort studies warrant to verify our findings.

Our findings demonstrate an intertwined nature of SDoH and CVH and their combined effects on mortality risks, which have implications for incorporating both SDoH and CVH into clinical practice and public health strategies to mitigate mortality disparities. Clinically, systematic screening for concurrent adverse SDoH and suboptimal CVH should identify high-risk individuals by healthcare providers and clinicians. Targeted interventions can then be deployed through multidisciplinary care teams addressing adverse SDoH while optimizing CVH. Concurrently, public health initiatives must address structural determinants through neighborhood investments, health education programs, and economic policies, which should be with population-level CVH promotion. Furthermore, combined assessments of SDoH and CVH are needed to enhance mortality risk prediction and stratification.

The strengths of the study included a nationally representative sample of U.S. adults and the systematic collection of SDoH and CVH metrics using standardized methods. However, several limitations should be acknowledged. First, causality cannot be inferred because of the observational design. Second, single timepoint assessment precludes examining how changes in SDoH or CVH over time influence mortality risk, which represents an important direction for future cohort studies. Third, residual confounding from unmeasured factors (e.g., genetic predisposition, medication adherence) may still exist. Fourth, behavior components of CVH (diet, physical activity, sleep, smoking) and several SDoH domains relied on self-report, which introduced recall bias, social desirability bias, and misclassification. Lastly, approximately 40% of participants were excluded due to missing data on CVH or SDoH components. Differences between the included and excluded participants may introduce selection bias, potentially limiting the generalizability of our findings. Thus, the results should be cautiously interpreted as representative primarily of U.S. adults with complete data on these measures.

## Conclusions

In a nationally representative sample of U.S. adults, the coexistence of a high burden of unfavorable SDoH and low CVH was associated with increased risks of all-cause, CVD, and cancer mortality. These findings provide implications for integrating both SDoH and CVH in clinical practice and public health strategies to improve survival outcomes.

## Supporting information

**S1 Fig. Flowchart of the study.**
(DOCX)

**S1 Table. Methods for evaluating each single cardiovascular health metric.**
(DOCX)

**S2 Table. Definitions of social determinants of health domains and sub-items.**
(DOCX)

**S3 Table. Baseline characteristics of participants included or excluded from the current analysis.**
(DOCX)

**S4 Table. Association between social determinants of health and cardiovascular health.**
(DOCX)

**S2 Fig. Mediation effects of cardiovascular health on the association of social determinants of health with mortality.**
(DOCX)

**S5 Table. Predictive value of models without and with the social determinants of health.**
(DOCX)

**S3 Fig. Association of cardiovascular health with all-cause and cause-specific mortality stratified by levels of social determinants of health.**
(DOCX)

**S6 Table. Characteristics of participants by combined categories of social determinants of health and cardiovascular health.**
(DOCX)

**S7 Table. Combined association of social determinants of health and cardiovascular health with all-cause and cause-specific mortality among US adults: subgroup analysis.**
(DOCX)

**S8 Table. Combined associations of social determinants of health and cardiovascular health with all-cause and cause-specific mortality among US adults after excluding participants died within 2 years of follow-up: sensitivity analysis.**
(DOCX)

**S9 Table. Combined associations of social determinants of health and cardiovascular health with all-cause and cause-specific mortality among US adults after excluding participants with cardiovascular diseases or cancer: sensitivity analysis.**
(DOCX)

**S10 Table. Combined associations of social determinants of health and cardiovascular health with all-cause and cause-specific mortality among US adults adjusted for more covariates: sensitivity analysis.**
(DOCX)

## Acknowledgments

The authors thank the investigators and participants of the NAHNES study.

## Author contributions

**Conceptualization:** Boxuan Pu, Zun Wang.

**Data curation:** Boxuan Pu.

**Formal analysis:** Boxuan Pu.

**Methodology:** Boxuan Pu.

**Supervision:** Zun Wang.

**Writing – original draft:** Boxuan Pu.

**Writing – review & editing:** Zun Wang.

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
