## [Decision Letter · Decision Letter 0]

7 Aug 2025

Dear Dr. Pu,

Thank you for submitting your manuscript to PLOS ONE. After careful consideration, we feel that it has merit but does not fully meet PLOS ONE’s publication criteria as it currently stands. Therefore, we invite you to submit a revised version of the manuscript that addresses the points raised during the review process.

Based on the reviewers' comments, there are some methodological concerns that we kindly ask you to address  in specific to enhance the clarity and soundness of your manuscript.

We look forward to receiving your revised manuscript.

Kind regards,

Angela Mendes Freitas

Academic Editor

PLOS ONE

Journal Requirements:

https://bmcpsychiatry.biomedcentral.com/articles/10.1186/s12888-024-06159-3

https://www.sciencedirect.com/science/article/abs/pii/S0022399924004446?via%3Dihub

In your revision ensure you cite all your sources (including your own works), and quote or rephrase any duplicated text outside the methods section. Further consideration is dependent on these concerns being addressed.

4. Please include captions for your Supporting Information files at the end of your manuscript, and update any in-text citations to match accordingly. Please see our Supporting Information guidelines for more information: http://journals.plos.org/plosone/s/supporting-information .

Reviewers' comments:

Reviewer's Responses to Questions

**Comments to the Author**

1. Is the manuscript technically sound, and do the data support the conclusions?

Reviewer #1: Yes

Reviewer #2: Yes

2. Has the statistical analysis been performed appropriately and rigorously?

Reviewer #1: Yes

Reviewer #2: No

3. Have the authors made all data underlying the findings in their manuscript fully available?

Reviewer #1: Yes

Reviewer #2: Yes

4. Is the manuscript presented in an intelligible fashion and written in standard English?

Reviewer #1: Yes

Reviewer #2: Yes

Reviewer #1: The text presented analysis the associations of cardiovascular health and social determinants of health with the risks of

all-cause and cause-specific mortality.

The text is very well organized and clearly written. The authors should be commended for their work and I have only suggestions to improve the way the paper communicates its message

Abstract:

The moderation (interaction) and mediation effects should be stated in the method part

Introduction

-the first time CVH is mentioned use the full form

Results

- I suggest the authors to consider other methods to measure mediation or to present measures such as direct effect| average mediated effect | indirect effect

- Similarly, I suggest a visual result scheme synthesis with the pathways proposed. This is more useful to better understand mediated and interaction analysis

Discussion

- The interpretation of the pathways could be better explained. For instance, "The interaction between SDoH and CVH on mortality was evaluated, revealing a significant additive interaction between these factors on all-cause mortality. Approximately one-eighth of all-cause mortality in individuals with low CVH and a high burden of unfavorable SDoH were attributed to the combined effects of these two risk factors"

But the authors do not explain how from the social and/or biological perspective there is an interaction

Reviewer #2: This study evaluates the combined impact of cardiovascular health (CVH), assessed by Life’s Essential 8 (LE8), and unfavorable social determinants of health (SDoH) on mortality risks among 20,096 U.S. adults using NHANES 2007-2018 data. The authors report notably elevated hazards for all-cause and cause-specific mortality in those with low CVH and ≥3 unfavorable SDoH compared to counterparts with high CVH and fewer SDoH issues (HR 4.60, 95% CI 2.87-7.37). The topic is highly relevant, and the dataset is robust. Here are my few comments and suggestions:

• Only 3.9% of the SDoH-mortality relationship is mediated by LE8, yet the manuscript strongly emphasizes CVH as a crucial mediator. Clearly address and contextualize this minor mediating role. Consider discussing other potential pathways or confounding factors explaining the observed associations.

• One of the major things that bothered me were missing covariates such as chronic kidney dx, lung dx, alcohol consumption, smoking, etc. I believe this could result in substantial residual confounding. I would suggest authors to perform additional analyses by including these important covariates.

• The manuscript used cause-specific cox models, treating deaths from other causes as censored observations. It would be better to do a competing risk analysis such as Fine-Gray competing risk to validate the robustness of the findings.

• Although understandable, many LE8 components such as diet, physical activity, sleep, smoking relied on self report single time assessments. I would encourage authors to highlight this database limitation and acknowledge the measurement bias that could have led to potential misclassification

• The authors excluded around 40% of NHANES respondents due to missing LE8 or SDoH data. I would like to see a more thorough comparison between included and excluded subjects and would encourage the authors to discuss implications for generalizability explicitly.

• The authors should discuss how their SDoH score compares with the scores used in previous SDoH and CVH studies:

o Satti DI, Chan JSK, Mszar R, Mehta A, Kwapong YA, Chan RNC, Agboola O, Spatz ES, Spitz JA, Nasir K, Javed Z, Bonomo JA, Sharma G. Social Determinants of Health, Cardiovascular Health, and Mortality in Sexual Minority Individuals in the United States. J Am Coll Cardiol. 2025 Feb 11;85(5):515-525. doi: 10.1016/j.jacc.2024.11.026. PMID: 39909683.

o Satti DI, Chan JSK, Dee EC, Lee YHA, Wai AKC, Dani SS, Virani SS, Shapiro MD, Sharma G, Liu T, Tse G. Associations Between Social Determinants of Health and Cardiovascular Health of U.S. Adult Cancer Survivors. JACC CardioOncol. 2023 Oct 24;6(3):439-450. doi: 10.1016/j.jaccao.2023.07.010. PMID: 38983373; PMCID: PMC11229543.

**Do you want your identity to be public for this peer review?** For information about this choice, including consent withdrawal, please see our Privacy Policy

Reviewer #1: No

Reviewer #2: No

---

## [Author Response · Author response to Decision Letter 1]

20 Sep 2025

Submission ID: PONE-D-25-13601

Associations of cardiovascular health and social determinants of health with the risks of all-cause and cause-specific mortality

Response to Comments

Thank you for inviting us to submit a revision of the manuscript for consideration for publication in PLOS One. We appreciate the editor and reviewers’ thoughtful comments, which have helped us to improve our submission. We have prepared responses and modified the manuscript based on the Editor’s requirements for resubmission and have responded to each comment. Specifically, herein we:

• Provide a point-by-point response to the reviewers' comments; the modified text, page numbers, paragraphs, and line numbers are provided.

• Itemize changes that have been made in the manuscript in response to comments by the reviewers.

• Provide two copies of the revision, one in which the changes made are highlighted and a clean second copy.

Reviewer Comments:

For Reviewer 1:

Reviewer #1: The text presented analysis the associations of cardiovascular health and social determinants of health with the risks of all-cause and cause-specific mortality.

The text is very well organized and clearly written. The authors should be commended for their work and I have only suggestions to improve the way the paper communicates its message

1. Abstract:

The moderation (interaction) and mediation effects should be stated in the method part

Response: Thanks for your suggestion. We have added the methods of mediation analysis and interaction analysis into the methods section of the abstract.

After revision:

Abstract (page 2, paragraph 2, lines 28-29): Mediation analysis was used to analyze the mediating role of CVH between SDoH and mortality. Interaction effects were tested by adding interaction terms between CVH and SDoH in the Cox models.

2. Introduction

-the first time CVH is mentioned use the full form

Response: Thank you for your reminder. We have already used the full name "cardiovascular health" (CVH) at the first occurrence in the main text of CVH, and subsequently, the abbreviation "CVH" is used in the rest of the main text.

3. Results

- I suggest the authors to consider other methods to measure mediation or to present measures such as direct effect| average mediated effect | indirect effect

- Similarly, I suggest a visual result scheme synthesis with the pathways proposed. This is more useful to better understand mediated and interaction analysis

Response: Thanks for your comments. We performed mediation analyses using the R package “mediation” to assess the extent to which CVH mediates the association between SDoH and mortality. We have now accounted for the complex survey design of NHANES by incorporating sampling weights through the R “survey” package prior to conducting mediation analysis. The “mediate” function was then applied using 1000 bootstrap simulations to ensure robust estimation of effect sizes and 95% confidence intervals. Accordingly, we have updated the description of the mediation analysis in the Methods section. The Results section and Table 3 have also been revised to present the updated mediation results, including the indirect (average causal mediation effect [ACME]), direct (average direct effect [ADE]), and total effects (ACME+ADE), as well as mediation proportions. Additionally, as suggested, we have added S2 Figure in the Supporting information to visually illustrate the mediation pathways and effect estimates.

After revision:

Methods (page 7, paragraph 4, lines 154-159): Mediation analyses were conducted using the R package "mediation" to quantify the extent to which CVH mediates the association between SDoH and mortality, with the proportion of mediation quantified by comparing models with and without adjustment for CVH. Bootstrap resampling (1000 simulations) was used to estimate of the mediation effect and its 95% confidence interval (95% CIs).

Results (page 9, paragraph 3, lines 212-218): When additionally adjusting for CVH, the HRs for the association between SDoH and all-cause, CVD, and cancer mortality were attenuated but still significant compared with models without CVH adjustment. Mediation analyses further showed that CVH significantly mediated the association between SDoH and all-cause mortality (mediation proportion: 16.70%; 95% CI: 10.90-18.96), CVD mortality (22.22%; 95% CI: 8.63-27.98), and cancer mortality (17.70%; 95% CI: 9.48-42.79) (Table 3, S2 Figure).

4.Discussion

The interpretation of the pathways could be better explained. For instance, "The interaction between SDoH and CVH on mortality was evaluated, revealing a significant additive interaction between these factors on all-cause mortality. Approximately one-eighth of all-cause mortality in individuals with low CVH and a high burden of unfavorable SDoH were attributed to the combined effects of these two risk factors" But the authors do not explain how from the social and/or biological perspective there is an interaction.

Response: Thank you for this insightful suggestion. We have revised the Discussion section to propose potential mechanisms of the interaction effect.

After revision:

Discussion (page 12, paragraph 2, lines 284-301): Moreover, we observed a significant additive interaction between SDoH and CVH on all-cause mortality. Approximately one-eighth of the excess mortality risk observed in individuals with both low CVH and high SDoH burden was attributable to synergistic effects. This suggests public health interventions simultaneously addressing both risk factors might yield greater mortality risk reduction than targeting either alone. Physiologically, adverse SDoH (e.g., poverty and limited healthcare access) may induce chronic physiological stress, such as dysregulated HPA axis and increased inflammation,7 which increases the adverse effects of low CVH. Resource constraints inherent to unfavorable SDoH concurrently limit the ability to promote CVH behaviors, such as accessing healthy food, safe environments for physical activity, and preventive care,10,23 thus creating a vicious cycle that accelerates health decline. However, the result of our study differs from a Chinese cohort study reporting no interaction between CVH and SDoH.27 This discrepancy may result from differences in SDoH and CVH definitions, as they used only 5 SDoH sub-items and did not use LE8 to assess CVH.27 Furthermore, it is noteworthy that high CVH provided greater protection against all-cause and CVD mortality in individuals facing SDoH disadvantages, suggesting the probability of optimizing CVH to mitigate mortality disparities exacerbated by adverse SDoH.

For Reviewer 2:

Reviewer #2: This study evaluates the combined impact of cardiovascular health (CVH), assessed by Life’s Essential 8 (LE8), and unfavorable social determinants of health (SDoH) on mortality risks among 20,096 U.S. adults using NHANES 2007-2018 data. The authors report notably elevated hazards for all-cause and cause-specific mortality in those with low CVH and ≥3 unfavorable SDoH compared to counterparts with high CVH and fewer SDoH issues (HR 4.60, 95% CI 2.87-7.37). The topic is highly relevant, and the dataset is robust. Here are my few comments and suggestions:

1. Only 3.9% of the SDoH-mortality relationship is mediated by LE8, yet the manuscript strongly emphasizes CVH as a crucial mediator. Clearly address and contextualize this minor mediating role. Consider discussing other potential pathways or confounding factors explaining the observed associations.

Response: Thanks for your comments. Our initial mediation analysis did not account for the complex survey design of NHANES. In this revision, we have re-conducted the analysis incorporating sampling weights and now report that CVH served as a significant mediator in the associations between SDoH and all-cause, cardiovascular, and cancer mortality, albeit with relatively modest mediation proportions. We agree that the magnitude of mediation is limited and have revised the manuscript accordingly to update these findings. In the Results section (Page 9, Line 214-218), we now state: "Mediation analyses further showed that CVH significantly mediated the association between SDoH and all-cause mortality (mediation proportion: 16.70%; 95% CI: 10.90-18.96), CVD mortality (22.22%; 95% CI: 8.63-27.98), and cancer mortality (17.70%; 95% CI: 9.48-42.79) (Table 3, S2 Figure)." to accurately show the magnitude.

Moreover, we have expanded the Discussion (Page 11, Lines 276-183) to discuss the modest mediating role: "Following adjustment for CVH, we observed an attenuation in the association between SDoH and mortality, suggesting a potential mediating role of CVH. Formal mediation analysis confirmed that CVH serves as a statistically significant but modest mediator in the pathways linking SDoH to both all-cause and cause-specific mortality. Nevertheless, the limited magnitude of mediation indicates that substantial reductions of adverse SDoH could not be achieved through promoting CVH alone, and other measures to tackle SDoH are still needed."

2. One of the major things that bothered me were missing covariates such as chronic kidney dx, lung dx, alcohol consumption, smoking, etc. I believe this could result in substantial residual confounding. I would suggest authors to perform additional analyses by including these important covariates.

Response: Thanks for your suggestions. 1. Smoking: as smoking status is a core component of CVH (health behavior domain), adjusting for it again das a separate covariate would constitute over-adjustment. Therefore, we did not include it as an additional covariate. 2. We agree these are potential confounders not originally adjusted for. We have now performed a sensitivity analysis further adjusting for additional covariates including drinking status, chronic kidney disease (CKD), and chronic obstructive pulmonary disease (COPD). Drinking status was classi�ed as current drinkers or noncurrent-drinkers. CKD was defined as meeting one of the following: estimated glomerular filtration rate (eGFR) <60 mL/min/1.73 m2 or urinary albumin-to-creatinine ratio (ACR)≥ 30 mg/g. COPD was self-reported history of COPD. The result of this sensitivity analysis is shown in S10 Table of the supporting information, which is consistent with the primary results.

3. The manuscript used cause-specific cox models, treating deaths from other causes as censored observations. It would be better to do a competing risk analysis such as Fine-Gray competing risk to validate the robustness of the findings.

Response: This is a good question. We acknowledge the theoretical advantage of competing risk models like Fine-Gray for cause-specific mortality. However, standard statistical software (e.g., R packages survey, cmprsk, tidycmprsk) currently lacks methods for incorporating NHANES sampling weights into Fine-Gray models. Using an unweighted Fine-Gray model might underestimate the nationally representative nature of our estimates. Therefore, to maintain the integrity of the population-representative analysis, we used the standard approach for cause-specific mortality in complex surveys: weighted Cox proportional hazards models treating deaths from other causes as censoring events. To address your concern about robustness, we performed unweighted Fine-Gray competing risk analyses of CVD mortality and cancer mortality (seen below). The results of CVD and cancer mortality were consistent with our primary weighted Cox models.

Outcomes CVH Hazard ratio (95% confidence interval)

CVD mortality

High burden of unfavorable SDoH High 1 (Reference)

Moderate 0.87 (0.49-2.32)

Low 1.62 (0.73-3.61)

Low burden of unfavorable SDoH High 0.34 (0.04-2.78)

Moderate 1.98 (0.92-4.26)

Low 2.38 (1.10-5.14)

Cancer mortality

High burden of unfavorable SDoH High 1 (Reference)

Moderate 1.95 (0.81-4.69)

Low 2.25 (0.91-5.57)

Low burden of unfavorable SDoH High 2.74 (0.85-8.87)

Moderate 2.46 (1.02-5.91)

Low 3.77 (1.56-9.08)

4. Although understandable, many LE8 components such as diet, physical activity, sleep, smoking relied on self-report single time assessments. I would encourage authors to highlight this database limitation and acknowledge the measurement bias that could have led to potential misclassification.

Response: We fully agree with the comment and have strengthened the acknowledgment of this limitation in the Discussion section (Page 13, Lines 334-346): "However, several limitations should be acknowledged. First, causality cannot be inferred because of the observational design. Second, single timepoint assessment precludes examining how changes in SDoH or CVH over time influence mortality risk, which represents an important direction for future cohort studies. Third, residual confounding from unmeasured factors (e.g., genetic predisposition, medication adherence) may still exist. Fourth, behavior components of CVH (diet, physical activity, sleep, smoking) and several SDoH domains relied on self-report, which introduced recall bias, social desirability bias, and misclassification. Lastly, approximately 40% of participants were excluded due to missing data on CVH or SDoH components. Differences between the included and excluded participants may introduce selection bias, potentially limiting the generalizability of our findings. Thus, the results should be cautiously interpreted as representative primarily of U.S. adults with complete data on these measures."

5. The authors excluded around 40% of NHANES respondents due to missing LE8 or SDoH data. I would like to see a more thorough comparison between included and excluded subjects and would encourage the authors to discuss implications for generalizability explicitly.

Response: Thanks for your comment. As shown in Figure S1 of the supporting information, the NHANES 2007-2018 database included a total of 59,842 participants. We first excluded 25,072 participants <20 years, 372 pregnant women, and 115 with missing values of mortality data. Then, we excluded 14,187 participants who lacked data on CVH or SDoH. Finally, we included 20,096 participants for analysis. Therefore, the proportion of participants with missing CVH or SDoH data was 35.7% of the total excluded participants. According to your suggestion, we added more characteristics in S3 Table of the supporting information to compare the baseline characteristics of participants included or excluded from our analysis. We found that excluded participants were younger, had a lower prevalence of CVD and cancer, and had more unfavorable SDoH than included participants; however, their CVH was comparable.

In addition, we have revised this limitation in the discussion "approximately 40% of participants were excluded due to missing data on CVH or SDoH components. Differences between the included and excluded participants may introduce selection bias, potentially limiting the generalizability of our findings. Thus, the results should be cautiously interpreted as representative primarily of U.S. adults with complete data on these measures.”

6. The authors should discuss how their SDoH score compares with the scores used in previous SDoH and CVH studies:

o Satti DI, Chan JSK, Mszar R, Mehta A, Kwapong YA, Chan RNC, Agboola O, Spatz ES, Spitz JA, Nasir K, Javed Z, Bonomo JA, Sharma G. Social Determinants of Health, Cardiovascular Health, and Mortality in Sexual Minority Individuals in the United States. J Am Coll Cardiol. 2025 Feb 11;85(5):515-525. doi: 10.1016/j.jacc.2024.11.026. PMID: 39909683.

o Satti DI, Chan JSK, Dee EC, Lee YHA, Wai AKC, Dani SS, Virani SS, Shapiro MD, Sharma G, Liu T, Tse G. Associations Between Social Determinants of Health and Cardiovascular Health of U.S. Adult Cancer Survivors. JACC CardioOncol. 2023 Oct 24;6(3):439-450. doi: 10.1016/j.jaccao.2023.07.010. PMID: 38983373; PMCID: PMC11229543.

Response: Thanks for your suggestion. We carefully read these two articles you mentioned. The two articles were both based on the same database "National Health Interview Survey (NHIS)", and were written by the same author team. Due to the limitation of the availability of NHIS survey data from the U.S., they used 6 domains to define SDoH; while we used 5 domains to define SDoH, compared with their domains, we lacked the "community and s

---

## [Decision Letter · Decision Letter 1]

7 Nov 2025

Associations of cardiovascular health and social determinants of health with the risks of all-cause and cause-specific mortality

PONE-D-25-13601R1

Dear Dr. Pu,

We’re pleased to inform you that your manuscript has been judged scientifically suitable for publication and will be formally accepted for publication once it meets all outstanding technical requirements.

Kind regards,

Angela Mendes Freitas

Academic Editor

PLOS ONE

Additional Editor Comments (optional):

Reviewers' comments:

Reviewer's Responses to Questions

**Comments to the Author**

Reviewer #1: All comments have been addressed

Reviewer #2: All comments have been addressed

2. Is the manuscript technically sound, and do the data support the conclusions?

Reviewer #1: Yes

Reviewer #2: Yes

3. Has the statistical analysis been performed appropriately and rigorously?

Reviewer #1: Yes

Reviewer #2: Yes

4. Have the authors made all data underlying the findings in their manuscript fully available?

Reviewer #1: Yes

Reviewer #2: Yes

5. Is the manuscript presented in an intelligible fashion and written in standard English?

Reviewer #1: Yes

Reviewer #2: Yes

Reviewer #1: The authors addressed all questions raised. I consider the text has the quality needed for publishing.

Reviewer #2: The manuscript has improved significantly and I commend the authorship team for making all the changes. I have no further comments.

**Do you want your identity to be public for this peer review?** For information about this choice, including consent withdrawal, please see our Privacy Policy

Reviewer #1: No

Reviewer #2: No

---

## [Editor Report · Acceptance letter]

PONE-D-25-13601R1

PLOS ONE

Dear Dr. Pu,

I'm pleased to inform you that your manuscript has been deemed suitable for publication in PLOS ONE. Congratulations! Your manuscript is now being handed over to our production team.

Kind regards,

on behalf of

Dr. Angela Mendes Freitas

Academic Editor

PLOS ONE